# Progress in the Preparation and Application of Breathable Membranes

**DOI:** 10.3390/polym16121686

**Published:** 2024-06-13

**Authors:** Tingshuai Luo, Ambar Farooq, Wenwei Weng, Shengchang Lu, Gai Luo, Hui Zhang, Jianguo Li, Xiaxing Zhou, Xiaobiao Wu, Liulian Huang, Lihui Chen, Hui Wu

**Affiliations:** 1College of Material Engineering, Fujian Agriculture and Forestry University, Fuzhou 350108, China; m15775107172@163.com (T.L.); ambarfarooq987@gmail.com (A.F.); zhangh10@163.com (H.Z.); jianguolicn@fafu.edu.cn (J.L.); star11110818@163.com (X.Z.); hll65212@163.com (L.H.); fafuclh@163.com (L.C.); 2Fujian Key Laboratory of Disposable Sanitary Products, Fujian Hengan International Group Company Ltd., Jinjiang 362261, China; wengww@hengan.com (W.W.); luog@hengan.com (G.L.); 3National Forestry and Grassland Administration Key Laboratory of Plant Fiber Functional Materials, Fuzhou 350108, China

**Keywords:** breathable, hydrophobic, membrane, micropore, electrospinning, application

## Abstract

Breathable membranes with micropores enable the transfer of gas molecules while blocking liquids and solids, and have a wide range of applications in medical, industrial, environmental, and energy fields. Breathability is highly influenced by the nature of a material, pore size, and pore structure. Preparation methods and the incorporation of functional materials are responsible for the variety of physical properties and applications of breathable membranes. In this review, the preparation methods of breathable membranes, including blown film extrusion, cast film extrusion, phase separation, and electrospinning, are discussed. According to the antibacterial, hydrophobic, thermal insulation, conductive, and adsorption properties, the application of breathable membranes in the fields of electronics, medicine, textiles, packaging, energy, and the environment are summarized. Perspectives on the development trends and challenges of breathable membranes are discussed.

## 1. Introduction

As membranes with microporous structures, breathable membranes [1,2,3,4,5] effectively regulate the passage of gas and water vapors. They block larger particles and water, offering the benefits of breathability and waterproofing, which has a wide range of applications, including medical [6,7,8], electronics [9,10], textile [11,12,13], packaging [14,15,16,17], energy [18,19,20], and environmental [21,22]. The key to their functionality lies in the size disparity between water droplets (0.2–3 mm) and water vapor molecules (0.4 nm). The pore size of breathable membranes is carefully calibrated to be smaller than water droplets yet larger than water vapor molecules. The intricate network of interconnected pores and channels ensures effective waterproofing while maintaining breathability.

Driven by the demand for comfort, health, and convenience, high-performance, innovative breathable membrane materials have been developed that offer great potential for multifunctional applications [23]. The breathability of a membrane is influenced by the material composition and the design of the porous structure [24,25,26]. In addition, by improving the manufacturing technology and modifying the treatment, multifunctional breathable membranes can be achieved [27,28]. Modifying the internal structure and incorporating adsorbents can make the breathable membrane effective to be used as air filter. Moreover, the integration of antimicrobial agents and thermal insulation materials allows the membranes to serve as medical dressings and warm, breathable clothing. Furthermore, high-performance waterproof breathable membranes have various applications in electronics [9], packaging [15], batteries [18], and desalination [29], exhibiting versatility and wide-ranging utility.

The overall objective of this review is to provide a comprehensive summary of the preparation, properties, and applications of breathable membranes. The preparation mechanism of breathable membranes, such as blown film extrusion, cast film extrusion, phase separation, and electrospinning, are explored. Keeping in view the antibacterial, hydrophobic, thermal insulation, conductivity, and adsorption properties, this study highlights the diverse applications of breathable membranes in the fields of medicine, electronics, textile, packaging, the environment, and energy. (Figure 1). The future challenges and prospects of breathable membranes are discussed to mount the way toward rational purposes and applications.

## 2. Preparation of Breathable Membranes

Breathable membranes can be produced through various strategies, such as blown film extrusion [30], cast film extrusion [31], phase separation [32,33], electrospinning [28], melt extrusion [5], and water droplet templating [34], depending on the material types and intended uses. Each fabrication technique plays a crucial role in determining the structure, functionality, and application of breathable membranes, influencing the membranes’ properties and performance significantly. Thus, in this section, four primary manufacturing approaches, including blown film extrusion, cast film extrusion, phase separation, and electrospinning, are discussed.

### 2.1. Blown Film Extrusion

Blown film extrusion [30,35] is a widely used technique for producing breathable membranes. In this process, molten plastic or resins with inorganic particles are fed through a screw into the head area of a film blowing machine and subsequently blown and cooled to form a continuous film [36]. To enhance breathability, the film undergoes machine direction orientation (MDO), where it is stretched at a specific temperature and ratio. The stretching separates the filler particles from the matrix, creating micropores. The MDO process includes four main steps: preheating, stretching, annealing, and cooling [37]. First, the film is heated to a determined temperature (60–75% of the range between the glass transition temperature and the melting point). Next, it is stretched between rolls at controlled speeds and temperatures, with the fast roll moving 2.5 to 10 times quicker than the slow roll, ensuring continuous stretching. The next step is the annealing hot roll, which achieves stress relaxation by holding the film at a high temperature for a period of time—this step provides dimensional stability—and finally, it is cooled at room temperature prior to winding [38]. Blown film extrusion allows for precise control over film thickness and air permeability by adjusting various parameters like extruder temperature, screw speed, blow ratio, and draw ratio, which produce effects on the film’s mechanical [39] and optical properties [40]. While blown film extrusion offers a high production rate and continuous operation, it requires high-temperature processing, leading to significant energy consumption.

### 2.2. Cast Film Extrusion

The cast film extrusion technique for creating breathable membranes involves a series of steps: melting the polymer with particles, casting them into a film, stretching the film to introduce micropores, and then cooling it down [31,41]. Micropores within the film are generated by stretching films that contain filled particles or lamellar crystals [42]. During the stretching process, when the film contains filled particles, a separation occurs between the polymer matrix and the filler particles. The separation leads to the formation of interconnected micropores around the inorganic particles, ensuring the polymer remains largely crack-free. Cast film extrusion is effective for producing membranes with specific porosity and breathability, without compromising the structural integrity of the polymer [43].

For some semi-crystalline polymers to form micropores by cast film extrusion, the polymer is first cooled and crystallized at a high temperature by extrusion and casting in the tensile stress field to form a hard elastic film [44,45] with parallel aligned lamellar crystals perpendicular to the direction of extrusion and an amorphous structure with good mobility. When the hard elastic polymer is annealed and extended in the orientation direction, the crystalline lamellae that are not connected by tie chains move apart, which leads to the bending of the crystalline lamellae and the formation of micropores between the crystalline lamellae and the amorphous polymer [42]. For hard elastic films, the higher the elastic recovery rates, the higher the degree of hard elastic structure formation, and the better the hard elasticity, the easier it is to stretch into pores, and the higher the porosity and permeability [46]. Adjusting the conditions of the casting process [47] can change the crystallinity, lamellar crystal thickness, and lamellar crystal orientation, thus adjusting the structure and air permeability of the breathable membranes. The draw ratio adjusts the degree of lamellar orientation, and the temperature of the casting roll controls the crystallinity and lamellar thickness. In addition, heat treatment or adding a certain number of other polymers can also improve the orientation of the crystals [48,49]. By precisely controlling the casting and stretching parameters, it is possible to fine-tune the performance of the breathable microporous film to suit various applications.

### 2.3. Phase Separation

Phase separation film formation techniques mainly include nonsolvent-induced phase separation (NIPS) [50,51,52] and thermally induced phase separation (TIPS) [32,53,54]. The NIPS process involves dissolving the polymer in a solvent to create a uniform casting solution. This solution is then shaped into a film and immersed in a nonsolvent for the polymer, typically water. The immersion triggers a solvent/nonsolvent exchange, leading to liquid–liquid phase separation and the formation of microporous membranes with various pore structures [50,55]. The amount of nonsolvent in the casting solution, the evaporation temperature, the evaporation time, and the solidification temperature have a great influence on the pore structure and the permeability of the membrane [50,56,57], this further affects the structure and performance of the membrane, for example, incorporating nonsolvent additives into the casting solution can enhance porosity and gas permeability. Evaporation at a low temperature leads to robust membranes with high gas permeability. Extending the evaporation time of the wet membranes increases the mechanical strength and air permeability. Lower coagulation temperatures also contribute to higher gas permeability. By carefully controlling the parameters, it is possible to tailor the membrane’s properties to meet specific requirements, allowing for customization in applications requiring precise permeability and structural characteristics.

The TIPS method [53] involves blending a high-melting-point polymer with a low-molecular-weight diluent, either liquid or solid. The mixture is then melted under heat and pressure. Phase separation is subsequently induced by cooling, with the diluent being removed through evaporation or other methods to form microporous structures. The diluent significantly impacts the phase separation mechanism and the resulting microporous structure. Microporous PVDF hollow fiber membranes can be prepared by the TIPS method using the environment-friendly hydrophobic solvent acetyl tributyl citrate (ATBC) [54,58], and the tensile temperature and the tensile ratio have an important effect on the fiber permeability, the tensile strength, the porosity, and the pore size. One of the advantages of the phase separation method for preparing membranes is that it does not require specialized equipment, leading to lower energy consumption [50]. However, environmental concerns arise due to the use of some solvents that are not eco-friendly, contributing to pollution. This challenge has led to an increased interest in green solvents that are environmentally safe and do not compromise the performance of the membranes. This shift towards sustainability reflects a broader trend in materials science to minimize environmental impact without sacrificing material quality or functionality.

### 2.4. Electrospinning

Electrospinning [59,60,61] utilizes electrostatic forces to draw a solution or melt, such as a polymer, from a tip or nozzle, creating a nanofiber film characterized by a high specific surface area and high porosity [62]. As a technique for generating nanofiber films, electrospinning offers lightweight material, small pore sizes, high porosity, and a customizable pore structure, which are ideal for creating a range of high-performance waterproof and breathable membranes [28,63]. Nanofiber breathable membranes have been developed using various materials, including polyurethane (PU) [64], polyacrylonitrile (PAN) [65], polyamide [23], and PVDF [66,67]. The properties of the electrospun membranes, such as pore size, porosity, and thickness, can be finely adjusted. This is achieved by modifying the spinning solution’s viscosity, electrical conductivity, surface tension, molecular weight, and concentration, as well as process parameters like feed rate, collector shape, voltage, tip-to-collector distance, and applied electric field. In addition, at the molecular level, some functional groups are modified to give the electrospun membranes superior properties [68,69]. These adjustments allow for tailored water repellency and air permeability in the final membranes. While electrospinning is effective for producing materials with fine fiber diameters and high porosity [70], it can sometimes result in compromised mechanical properties, due to the instability of the jet during the spinning process [71]. Nevertheless, the ability to produce breathable membranes with such high specific surface areas and porosities makes electrospinning a promising method in the development of advanced functional materials.

As displayed in Table 1, various preparation methods for breathable membranes feature different pore formation principles. Blown film stretching and cast film extrusion can both form pores by stretching to separate the matrix from the filled particles; phase separation forms pores by separating the different component phases and then removing the solvent or diluent; and electrospinning forms pores by the fibers that are formed intertwining with each other. The blown film stretching and cast film extrusion preparation processes will not cause harmful effects on the environment, but in the phase separation and electrospinning processes, there will be an impact on the environment due to the need for organic solvents for the treatment of solutions or polymers. Thus, it is crucial to develop environmentally friendly solvents or new technological approaches, aiming to eliminate/reduce environmental pollution. Optimizing the preparation method of breathable membranes to further expand their applications is an important direction in the research and development of breathable membranes.

## 3. Applications of Breathable Membranes

Breathable membranes excel in providing waterproofing and breathability. By integrating specific functional groups or substances, the membranes can possess multifaceted capabilities, including antimicrobial, thermal insulation, electrical conductivity, and adsorption properties. Below is a summary of the application of breathable membranes in the fields of medicine, electronics, textile, packaging, and the environment.

### 3.1. Medical Field

Breathable membranes can be prepared as electronic sensors for medical detection applications due to their good breathability [27,74,75,76,77,78]. Yan et al. [1] developed an approach for wearable electronics, addressing the challenge of integrating electronic functionality with the flexibility and breathability required for medical detection applications. Molybdenum disulfide (MoS_2_) nanosheet ink was synthesized through an intercalation/exfoliation method and a spin-coating process was used to assemble the nanosheets into van der Waals thin-film electronics (VDWTF) (Figure 2a). This method resulted in electronics that are highly stretchable, adaptive, conformal, and breathable, offering an excellent match to the mechanical and physiological characteristics of soft biological tissues. Traditional hard electronic materials often fail to align with the soft, flexible nature of biological tissues in terms of electrical conductivity, mechanical response, permeability, and environmental adaptability. The VDWTFs overcome the limitations through the unique structure. The non-bonded van der Waals interfaces between the MoS_2_ nanosheets allow for sliding and rotation, granting the membrane extraordinary mechanical flexibility and ductility. This capability enables the VDWTFs to accommodate localized stretching or compression without forming cracks or breaks (Figure 2b), and the staggered nanosheet structure creates a permeable network of nanochannels, enhancing breathability (Figure 2c). The ultrathin VDWTFs are not only structurally robust but also exhibit a good mechanical match with biological soft tissues, allowing for direct bonding to living organisms through highly conformal interfaces. This feature enables the integration of electronic detection functions directly onto the skin. Skin-gate VDWTF transistors (Figure 2d) can be applied to human skin, offering a cutoff frequency of ~100 kHz (Figure 2e) and the capacity to monitor electrophysiological signals from the body. In practical applications such as ECG monitoring (Figure 2f), VDWTF electrodes have demonstrated stronger signal strength and lower signal noise compared to traditional Ag/AgCl electrodes. This advantage enables the VDWTF electrode to clearly identify critical components of the ECG waveform (“PQRST”) (Figure 2g), which is crucial for diagnosing cardiovascular conditions like myocardial infarction and arrhythmias [79]. Additionally, VDWTFs can record high-quality EEG signals (Figure 2h), revealing significant dynamics in alpha rhythms associated with eye opening and closing (Figure 2i). This capability is vital for monitoring brain activity and researching various neurological disorders. These membranes are not only capable of adapting to the environment but also retain the electronic properties for advanced sensing, signal amplification, processing, and communication tasks. The development of such breathable, stretchable, and conformal electronics marks a significant advancement in wearable technology, particularly for medical detection and monitoring applications.

Breathable membranes play a pivotal role in medical applications, notably in wound dressings [8,26,80], burns, and post-surgical care, as well as in protective clothing [81,82,83,84]. The membranes facilitate oxygen exchange and moisture regulation at the wound site, promote healing, and reduce infection risks [85]. The controlled moisture environment mitigates heat buildup, further aiding the healing process. The integration of antimicrobial agents into breathable membranes enhances the functionality, effectively inhibiting or eliminating bacteria to prevent infections. These agents can range from organic soluble antimicrobial substances and inorganic metal-based antimicrobials to natural materials like chitosan [86,87,88]. Such advancements significantly improve the membranes’ breathability and antimicrobial efficacy and enable breathable membranes to have good breathability and antimicrobial efficacy [89]. Yue et al.’s study [8] involves the use of a portable, gun-shaped electrostatic spinning device (Figure 3a) to create waterproof and breathable nanofiber membranes for wound dressings (Figure 3b). The membranes, made from fluorinated polyurethane (FPU) and ethanol-soluble polyurethane (EPU), were further enhanced with thymol, a natural antimicrobial agent (Figure 3c). The addition bestowed the membranes with antimicrobial properties, essential for preventing bacterial infections and minimizing wound irritation from external elements. The EPU/FPU nanofibrous membranes exhibited an intricate pore structure, demonstrating superior mechanical strength (tensile stress of 1.83 MPa) (Figure 3d) and hydrophobicity (WCA > 144°) (Figure 3e), compared to other materials like polyvinyl pyrrolidone (PVP) and polyvinyl butyral (PVB). Remarkably, these membranes exhibited excellent breathability (3.56 kg m^−2^d^−1^) (Figure 3f), as evidenced by the ability to allow water vapor to pass through effortlessly, proving their efficacy as wound dressing materials. This approach not only yields a membrane with good mechanical properties and breathability akin to human skin, but also incorporates antimicrobial protection through thymol. Such membranes offer a promising avenue for applications in wound dressings and potentially in flexible electronic sensors, highlighting the significant potential of breathable membranes in the medical field.

The COVID-19 pandemic has significantly heightened awareness and demand for effective virus prevention measures, particularly in the development of high-performance protective clothing [90]; the earliest use of protective clothing began in hospitals to prevent bacterial, viral, and other types of microbial contamination and to save patients from being infected by the viruses carried by medical personnel [91]. Ideal protective clothing must offer excellent bactericidal barrier properties without compromising breathability, waterproofing, and moisture permeability, ensuring comfort for medical staff during extended wear periods. Current commercial protective clothing materials, including breathable polyethylene film/nonwoven composite (SF), flash-spun nonwoven (FS), and spun bond/melt-blown/spun bond polypropylene (SMS) membranes, face challenges balancing filtration efficiency with breathability [90,92]. SF and FS materials, despite the high filtration efficiency and barrier properties, suffer from low porosity and poor breathability [93]; compared to SF and FS, SMS has high breathability due to its abundant large pores and fluffy structure, and SMS has an electrostatic effect on the carryover charge of particulate matter, thus providing barrier performance [94]. However, when SMS is exposed to water or disinfectant, the electrostatic effect is unstable, and the filtration efficiency may be drastically reduced by the dissipation of the electrostatic charge [95]. Therefore, high-performance protective clothing with both high barrier properties and high waterproofing, breathability, and moisture permeability is essential.

Li et al. [7] developed an innovative approach by using pyromellitic dianhydride (PMDA) and 4,4′-oxidianiline (ODA) to create a polyimide (PI) electrospun nanofiber membrane (PI-ENM) with high filtration efficiency, excellent breathability, and strong mechanical strength. The membrane was further enhanced by coating it with a thin layer of polydimethylsiloxane (PDMS) through chemical vapor deposition (CVD), resulting in a PI-ENM@PDMS membrane suitable for protective clothing (Figure 4a). The increasing fiber diameter from 200 nm to 500 nm improved the tensile strength, indicating superior mechanical property (Figure 4b). Filtration performance tests demonstrated that PI-ENMs could effectively filter ultra-fine particles (20 nm) with a 99.42% efficiency (Figure 4c), providing a high microbial barrier performance. Additionally, the waterproof and breathable properties were highlighted by the rapid color change of silica gel humidity indicators, confirming the membrane’s high permeability to water vapor (Figure 4d). The PI-ENM also exhibited an excellent waterproof performance (WCA > 135°) (Figure 4e), thermal stability (heat resistance > 200 °C) (Figure 4f), and a degree of flame retardancy, which could be further enhanced by the PDMS coating (Figure 4g). The study by Li et al. presents PI-ENM@PDMS as a promising material for the next generation of reusable protective clothing, offering an optimal balance of high air permeability, superior barrier performance, mechanical strength, thermal stability, and flame retardancy. This advancement represents a significant step forward in protective clothing technology, meeting the critical need for effective, comfortable, and durable protection against viral and microbial threats.

### 3.2. Electronics Field

Wearable electronics [96,97,98,99,100,101,102] can accurately detect human motions and physiological signals. Breathable membranes can be used in wearable electronic devices to collect multifunctional biomechanical energy by monitoring various human movements. Sun et al. [9] fabricated a breathable and waterproof wearable triboelectric nanogenerator (NF-TENG) with an innovative all-fiber structure (Figure 5a). The NF-TENG comprises three key components: a friction-positive layer made of a PA66/MWCNT nanofiber film, which is a blend of thermoplastic resin (PA66) and multi-walled carbon nanotubes (MWCNTs); a friction-negative layer consisting of a PVDF nanofiber film; and a layer combining conductive fabric with a waterproof and breathable fabric (Figure 5b). This structure capitalizes on the unique properties of the materials used, resulting in a device that is highly efficient at converting mechanical energy into electrical energy, while maintaining the essential qualities of breathability and waterproofness. The nano-mesh structure (Figure 5c) increases the surface roughness and specific surface area and improves the electrical properties of the NF-TENG. The friction-negative layer of the PVDF film consists of electrostatically spun nanofibers (Figure 5d). The synergy effect of contact electrification and electrostatic induction realized the energy collection of the NF-TENG (Figure 5e): the frictional electric pairs do not generate charge from the separated state, which results in no potential difference between the frictional electric layers, and when the frictional electric materials closely oscillate, a friction charging process occurs, and the frictional electric materials generate equivalent opposite charges at the interface, and a potential difference is still not formed [103,104]; with the release of the external pressure, a potential difference between the frictional electric materials is generated, which induces the free electrons to flow through the external loads from the PVDF films to PA66/MWCNT films through external loading, resulting in transient current flow. As the external force is gradually unloaded, the potential difference increases. Until the PA66/MWCNTs film is completely restored to its original position, the potential difference reaches its maximum value. When the external force approaches the PA66/MWCNTs film again, the original electrostatic equilibrium is broken, resulting in the electrons of the PA66/MWCNT film flowing back to the PVDF film. When the two sides are in complete contact, the potential difference becomes zero. Successive compression and release processes bring about pulses of voltage and current to convert mechanical energy into electrical energy. Moreover, the hydrophobic nature of the NF-TENG is demonstrated by the water contact angles (WCAs) of the PA66/MWCNT and PVDF films, measured at 120° and 135°, respectively (Figure 5f). This ensures that the device remains functional even in moist conditions. The NF-TENG also exhibits excellent breathability (Figure 5g), with air permeability rates surpassing those of commercial jeans and other wearable devices [105,106]. The breathable and waterproof wearable NF-TENGs can be placed at any body position to monitor various human movements and power wearable electronic devices for daily use (Figure 5h). The incorporation of MWCNTs into the PA66 film was found to decrease its tensile strength, resulting in more general mechanical properties. This indicates a trade-off between enhancing the electrical properties of the NF-TENG and maintaining superior mechanical strength. Nevertheless, the NF-TENG’s capability to monitor human movements and power wearable electronic devices opens possibilities for the integration of energy-harvesting devices into everyday wearables, offering a promising approach to powering electronic devices in a sustainable and user-friendly manner.

### 3.3. Textile Field

In the realms of apparel, outdoor equipment, and footwear, breathable membranes [107,108,109] serve the crucial function of preventing moisture ingress while allowing water vapor to escape, thus ensuring a comfortable wearing experience for the user. It is essential for stabilizing body temperature to increase the thermal insulation and warmth of textiles. To enhance the thermal insulation properties of breathable membranes, thereby improving heat retention, several heat-insulating materials are incorporated into the membranes. These materials include long-chain alkyl polymers [11], silica aerogels [12], and silver nanoparticles [13]. Each of these additions contributes uniquely to the membrane’s ability to minimize heat loss. Long-chain alkyl polymers are known for their low thermal conductivity, making them excellent insulators. When integrated into breathable membranes, they are added as a layer of insulation that helps trap body heat, reducing heat loss to the environment. Silica aerogels possess an extraordinary thermal insulation capacity due to their high porosity and low thermal conductivity. Their incorporation into breathable membranes enhances the membranes’ ability to retain heat, offering superior warmth without significantly increasing weight or bulk. Silver nanoparticles not only have antimicrobial properties but also reflect infrared radiation, which is a component of body heat. By reflecting the radiation back towards the body, silver nanoparticles contribute to the thermal insulation of the membrane, aiding in heat retention.

Zhou et al. [11] prepared fluorine-free hydrophobic nanofibrous membranes for protective textiles by in situ doping of long-chain alkyl polymers (LAP) and polycarbodiimide, (PCD) using electrospinning with water as the solvent (Figure 6a). A simple heat treatment was employed to enrich the textile surface with long-chain alkyl groups, leading to the creation of a nanofiber membrane characterized by an interconnected porous channel structure (Figure 6b). The structure, with tiny pore sizes and high porosity, effectively resists liquid water penetration while facilitating the easy transport of water vapor/air, showing remarkable waterproofing and breathability (Figure 6c). The hydrophobic properties of the membrane were directly correlated with the LAP content; as the amount of LAP increased, so did the presence of surface long-chain alkyl groups on the nanofibers, resulting in a notable increase in the water contact angle (WCA) measurements of the nanofiber membrane, recorded at 134.6°, 137.1°, and 139.2°, respectively. Moreover, the nanofiber membrane demonstrated excellent hydrophobicity against substances like milk and coffee (Figure 6d). The water vapor permeability of the nanofiber membranes under varying environmental conditions was also assessed. Tests revealed that the membranes offer enhanced breathability (4885 gm^−2^d-^1^) in conditions of higher temperature and lower humidity, particularly showing higher water vapor flow rates at a temperature of 43 °C and a humidity of 40% (Figure 6e). During tensile testing, the membrane displayed remarkable stretchability and resilience, easily recovering after being stretched to 300% tensile strain, with an impressive tensile elongation of 372.4% and elasticity of 56.9%, demonstrating its excellent mechanical properties (Figure 6f). The eco-friendly, fluorine-free nanofibrous membrane with waterproof properties and breathability, manufactured through a one-step waterborne electrospinning technique, has potential applications in medical hygiene, wearable electronics, seawater desalination, and oil/water separation.

Ideal cold weather clothing not only needs to protect against moisture and allow for breathability but also must have heat retention to keep the wearer warm. To address this, thermal insulation materials are integrated into breathable membranes to enhance the heat retention capabilities. Among these materials, silica aerogel stands out due to its exceptionally low thermal conductivity, making it a superb insulator. Shi et al. [12] developed polyurethane-based breathable membranes (THSPUs) embedded with silica aerogel (SA) particles, utilizing a blending technique (Figure 7a). These membranes effectively retain heat when the SA content is kept below 2.0%, as the SAs are uniformly dispersed within the THSPUs, creating a barrier against heat conduction (Figure 7b). However, it was noted that exceeding an SA content of 1.0% leads to particle agglomeration, which can compromise the membrane’s mechanical properties. As the content of SAs increases, the hydrophilicity of the THSPU membranes shifts towards hydrophobicity, with the water contact angle rising from 62.3° to 107.5°. Furthermore, the water vapor transmission rate (WVT) of the THSPU membranes is temperature-sensitive, enhancing air permeability adaptively within the human body’s comfortable temperature range as the temperature climbs. These membranes also boast commendable mechanical strength, evidenced by the ability to withstand hydrostatic pressure up to 17,800 mm H_2_O, exhibiting good durability. The THSPU membranes have the potential to be utilized in sports apparel, fashion rainwear, and protective workwear. This is due to their capability to manage personal thermal comfort effectively, even in harsh environmental conditions, by combining heat retention, durability, and waterproof–breathable properties.

Infrared (IR) radiation represents a significant pathway for heat loss from the human body, accounting for over 50% of bodily heat emission [110]. Traditional garments typically lack the capability to insulate against this form of heat loss effectively. Addressing this challenge, Yue et al. [13] developed a thermal insulation membrane consisting of silver nanoparticles and cellulose fibers (Ag-NPs/CTIM) that boasts high infrared reflectivity, excellent breathability, and robust antibacterial properties. The membrane was created using a straightforward silver mirror reaction and vacuum filtration process, with cellulose fibers derived from wastepaper serving as the base material. The integration of silver nanoparticles (Ag-NPs) into the cellulose fibers creates an efficient infrared radiation reflective layer. This layer serves to reflect the thermal radiation emitted by the human body, thereby diminishing heat loss and enhancing the thermal insulation performance of clothing. Such technology holds considerable promise for applications aimed at preserving human body warmth. Moreover, the Ag-NPs/CTIM membrane exhibits significant antibacterial activity. However, despite these advantages, the membrane’s inherent hydrophilicity poses a limitation. While its ability to absorb sweat may improve comfort under normal conditions, this feature could become a drawback in rainy weather. Absorbing water might reduce both the insulation capability and comfort level of the clothing. Thus, modifying the Ag-NPs/CTIM to possess hydrophobic qualities could enhance its utility, ensuring that it remains effective and comfortable in a wider range of environmental conditions, including wet weather. This adjustment would make the membrane more versatile, suitable for diverse applications where thermal management and comfort are crucial.

The above progress in breathable membrane technology significantly contributes to the development of textiles that not only provide protection against moisture and environmental elements but also ensure thermal comfort and efficiency. This makes them ideal for use in cold climates and in the design of high-performance outdoor and sports apparel, where the regulation of body temperature and moisture management are key to comfort and performance.

### 3.4. Packaging Field

Biodegradable polymers are the first choice of materials for producing environmentally friendly food packaging, but they usually have poor barrier properties and protective functions compared to petroleum-based conventional plastics, resulting in reduced food quality and shortened shelf life [111]. The development of biodegradable and environment-friendly food packaging materials [14,112,113] that can extend the shelf life of food products has important applications. Phothisarattana et al. [14] tackled this challenge by employing the blown extrusion method (Figure 8a) to create PBAT/TPS/ZnO nanofilms. ZnO nanoparticles were integrated into polybutylene adipate-co-terephthalate/thermoplastic starch (PBAT/TPS) films (Figure 8b) to enhance the properties. The PBAT/TPS films exhibited holes due to the incompatibility between the substances (Figure 8c). TiO_2_ and ZnO are the most common additives used for self-cleaning and antibacterial surfaces [114]. The addition of ZnO nanoparticles promotes the formation of nanopores so that the film contains more pore space (Figure 8d). Incorporating 3% and 4% ZnO into the films significantly enhanced the water vapor permeability (WVP), while the films containing 2% ZnO had the best hydrophobicity (Figure 8e). The modification not only improved the physical properties of the films but also extended the shelf life of meat products. Fresh pork stored in PBAT/TPS packaging without ZnO saw its total viable count (TVC) exceed acceptable limits (7 log cfu/g) [115] after 9 days, indicating spoilage. In contrast, films containing 1–5% ZnO effectively controlled the TVC below the threshold, demonstrating antimicrobial efficacy (Figure 8f). The biodegradable and environmentally friendly PBAT/TPS/ZnO nanofilms show promise in extending the shelf life of packaged foods. This aligns with the growing demand for sustainable packaging solutions, offering a viable alternative to conventional plastics while contributing to food preservation and waste reduction.

For fresh fruits and vegetables, the quality after picking is affected by a variety of factors such as respiration rate, moisture content, and storage environment [116]. An excessively high oxygen level in the storage environment accelerates aerobic respiration in the packaged produce, leading to a rapid loss of nutrients. Conversely, too low oxygen levels result in the accumulation of anaerobic metabolites, such as ethanol, which negatively affects the storage of freshly picked fruits and vegetables [117]. Therefore, microporous and breathable packaging materials can be used to regulate the gas transfer rate (GTR), which means maintaining the optimal equilibrium concentration of oxygen (O_2_) and carbon dioxide (CO_2_) inside food packaging to solve this problem [118,119]. Additionally, these materials help prevent the intrusion of insects and microorganisms and block water infiltration, thereby preserving the freshness and quality of the food products [17,120]. Wu et al. [15] developed an air-permeable packaging using a polymer of intrinsic microporosity (PIM-1) combined with polylactic acid (PLA). The resulting PIM-1/PLA membrane is breathable and selective, capable of balancing gas concentrations and facilitating water drainage, which in turn inhibits mango respiration and delays the deterioration of mango quality. To further enhance the protection of food during storage, the incorporation of antimicrobial agents into porous food packaging materials is critical. Cinnamaldehyde, a natural antimicrobial agent, was integrated into polymer membranes to bolster the antimicrobial effectiveness [121]. In a related study, breathable packaging films were created using poly (lactic acid) (PLA), poly(caprolactone) (PCL), and porogenic agents (sodium chloride and polyethylene oxide) through solvent casting [16]. The optimal composition found was 20% PLA and 80% PCL, with 50% sodium chloride and 10% poly (ethylene oxide), resulting in the highest air and oxygen permeability. The inclusion of cinnamaldehyde further enhanced the antimicrobial properties of the film, making it more suitable for food packaging applications. The use of PIM-1/PLA membranes not only supports the creation of an equilibrium gas concentration and efficient water discharge but also offers a method to inhibit the respiration and quality degradation of mangoes. This approach holds promise for extending to other fruits and vegetables, with future research aimed at exploring the broader applicability of PIM-1/PLA membranes in fruit storage. Furthermore, the incorporation of cinnamaldehyde into PCL/PLA membranes represents a significant step towards developing antimicrobial, breathable films from biodegradable polymers for food packaging, showing sustainability in preserving food quality.

### 3.5. Energy Field

Metal–air batteries have attracted much attention due to their low cost and high energy density [122]. However, the electrochemical performance of metal–air batteries is sensitive to ambient humidity, which severely limits their application in humid environments [123]. Installing a waterproof and breathable membrane in the battery can effectively solve the problem of the battery’s sensitivity to ambient humidity [124]. In addition, the microporous structure of the breathable film can also be used as an electrode to improve battery performance [125]. Wang et al. [18] prepared polydimethylsiloxane/polytetrafluoroethylene (PDMS/PTFE) composite membranes by using the water surface spreading method (Figure 9a). The study investigated the impact of spreading times and the addition of a hydrophobic SiO_2_ (HC-SiO_2_) filler on the membranes’ water vapor transmission rate and air permeability. The air permeability of the PDMS/PTFE membranes initially decreased and then increased with longer spreading times, whereas water vapor permeability showed an initial decrease followed by fluctuations (Figure 9b). With the increase in the HC-SiO_2_ mass fraction, the water vapor permeability of S-HC-SiO_2_/PDMS/PTFE membranes decreased first and then increased (Figure 9c), and the waterproof ability and air permeability of PDMS/PTFE membranes could be improved by increasing the spreading time and adding an appropriate amount of HC-SiO_2_. The S-HC-SiO_2_/PDMS/PTFE membrane was assembled into a lithium–air battery (Figure 9d). Without the protection of a waterproof and breathable membrane, the operation resistance of lithium–air batteries increased dramatically as the operating time increased; due to the formation of non-conductive LiOH, the lithium anode corroded rapidly in a 40% RH air environment, resulting in a rapid increase in the operating internal resistance. For the battery loaded with the S-HC-SiO_2_/PDMS/PPTFE film, the operating internal resistance increased slowly, and the battery exhibited superior performance (Figure 9e); the resistance of the lithium–air battery could have been reduced due to the protection of the lithium anode by the breathable membrane. The performance of the waterproof and breathable membrane was further evaluated by measuring the cycling stability of the lithium–air batteries. The lithium–air battery/PDMS/PTFE membrane equipped with S-HC-SiO_2_ could reach 25 cycles in an air environment (40% RH, 25 °C), and the charging and discharging polarization voltage only increased from 0.44 V to 0.75 V (Figure 9f), which is much better than that of the battery without a diaphragm whose cycle number was 20 (45 h), and the charge/discharge polarization voltage difference increased from 0.81 V to 1.75 V. Furthermore, the membranes also showed promising results in zinc–air batteries, effectively reducing electrolyte evaporation and battery resistance, thus enhancing cycling stability. The batteries with the membrane achieved more cycles and displayed more stable charging and discharging polarization voltages over long periods compared to those without the membrane. Despite the environmental concerns related to the use of organic solvents in the water surface spreading method, the benefits of the S-HC-SiO_2_/PDMS/PTFE membrane are clear. It not only improves the performance of lithium–air and zinc–air batteries in humid conditions by protecting against electrode corrosion, but also enables stable operation at lower charging and higher discharging voltages. The composite membranes significantly enhance the viability and durability of metal–air batteries, marking a step forward in the development of more reliable and environmentally friendly energy storage.

Amici et al. [19] fabricated a highly hydrophobic and oxygen-selective membrane, which significantly enhances the performance and lifespan of lithium–air batteries. The membrane was made by doping a hydrophobic polymer dextrin-nanosponge (NS) into a poly (vinylidene fluoride-co-hexafluoropropylene) (PVDF-HFP) matrix through a nonsolvent-induced phase transition method, showing remarkable improvements in both oxygen permeability and water repellency compared to traditional membranes. The study conducted a comparative analysis of the oxygen permeability and water repellency between the newly developed PVDF-HFP-NS membrane, a pristine PVDF-HFP membrane, and a silicone oil-loaded PVDF-HFF membrane. The pristine PVDF-HFP membrane exhibited very low oxygen permeability and water selectivity, which would drastically limit the amount of oxygen entering a cell, far below the required levels for optimal battery operation. On the other hand, the PVDF-HFP-NS membrane demonstrated a six-fold increase in oxygen permeability and significantly reduced water permeability compared to the PVDF-HFF membrane. Additionally, hydrophobicity tests revealed that the PVDF-HFP membrane has a contact angle (CA) of 125.4 ± 5.7°, indicating a high level of hydrophobicity, which underscores its ability to effectively block the ingress of water molecules while selectively allowing oxygen to pass through. The PVDF-HFP-NS membrane’s efficacy was further validated in lithium–air batteries, where batteries equipped with the membrane achieved 143 cycles at a certain relative humidity (RH), boasting an operational life of 1450 h. This performance starkly contrasts with batteries lacking the membrane, which managed only 37 cycles and an operational lifespan of 370 h. Such a substantial improvement not only underlines the membrane’s superior oxygen selectivity and hydrophobic nature but also its crucial role in enhancing battery reliability and longevity in humid conditions. Creating membranes that effectively balance breathability with waterproof capabilities has been a considerable hurdle in battery technology. The success of the PVDF-HFP-NS membrane demonstrates the potential for such materials to solve critical issues in battery performance and life cycle, marking a promising direction for future research and development in energy storage solutions.

### 3.6. Environmental Field

Air pollution, particularly fine particulate matter with diameters of 2.5 μm or smaller (PM_2.5_), poses significant risks to public health and the environment [126]. Long-term exposure to such particles can lead to severe health issues, including pneumonia, lung cancer, coronary artery disease, and congestive heart failure [127,128]. The situation is further exacerbated by irregular viral outbreaks, underscoring the critical need for efficient filtration devices capable of protecting against pathogenic microorganisms. Breathable membranes, with selective air permeability, emerge as a promising solution for addressing these challenges. These membranes can serve multiple roles, not only in air purification systems to separate and filter out contaminants like PM_2.5_ and harmful pathogens [21,129,130,131], but also in water treatment processes [132,133,134]. The ability to selectively allow certain molecules to pass through while blocking others makes them highly effective in improving air quality, safeguarding public health, and ensuring environmental protection. The conventional passive capture method of blocking pollutants through dense mesh nanofiber structures can exhibit interception effects in the presence of simple and common PM air streams, but the pressure drops and permeability do not perform well when intercepting finer PM at high air streams [95], so multilayer structures in nanofibers have been developed in order to balance the relationship between interception efficiency and pressure drop.

Deng et al. [21] developed sodium sulfobutyl ether-β-cyclodextrin/polyvinyl alcohol (SBE-βCD/PVA) nanofiber membranes through electrospinning. The membrane contains a multistage bent ribbon nanofiber structure, which contrasts with traditional rod-shaped fibers (Figure 10a,b). The unique design offers a morphological barrier that significantly enhances PM interception. The bent ribbon structure increases the contact area with airborne particles, preventing them from simply sliding off the fiber surface. Additionally, the incorporation of sulfobutyl polar functional groups from SBE-βCD into the fibers introduces a plethora of negatively charged electrostatic sites. These sites are crucial for capturing PM, markedly improving the membrane’s filtration efficiency [135]. The bent ribbon fibers outperform other nanofibers in filtration efficiency across all measured basis weights (Figure 10c). Remarkably, with a basis weight of 6 g/m^2^, the filtration efficiency for PM_1.0_ particles reached an impressive 99.12%. Despite the inherent challenge of increased filtration pressure drop associated with higher basis weights, the unique porous structure of the curved-ribbon fibers maintains the pressure drop at an acceptable level of 57.5 Pa^−1^ while preserving a high filtration efficiency (Figure 10d). Mechanical properties are crucial for ensuring the long-term stability and effectiveness of air filtration materials. The bent ribbon fibers exhibit superior tensile strength and strain compared to conventional rod-shaped fibers, with a strain reaching up to 120% (Figure 10e). The elasticity is advantageous for air filters, allowing them to maintain a stable and breathable structure even under significant external air resistance. The potential application of the electrostatically spun nanofiber membrane extends to protective masks (Figure 10f). By substituting the core filter layer of a common medical mask (C-mask) with the curved-ribbon nanofiber membrane, a reassembled mask (R-mask) was created. Comparative tests with common medical masks, KN95 masks, and the R-mask demonstrated that the R-mask achieved a filtration efficiency of 99.9% with a lower pressure drop of 59.5 Pa^−1^, offering near-perfect filtration for PM_1.0_ and PM_2.5_ (Figure 10g). While slightly less efficient than the KN95 mask, the R-mask presents a substantial improvement in wearer comfort due to its lower pressure drop, highlighting its potential as a more comfortable yet highly effective alternative for air filtration and personal protection. Using green one-step electrostatic spinning, curved-ribbon nanofiber membranes with multilayered loose structures are efficient and sustainable for air filtration. Due to the wide diameters, the ribbon fibers show potential advantages in easily intercepting particles, but they also block airflow, reducing air permeability. It is worth investigating how to construct self-curling nanofibers with smaller diameters to form filters with both small pores and high porosity to achieve simultaneous improvement in filtration efficiency and air permeability.

Cheng et al. [22] created ordered a PAN/nano-spiderweb composite nanofiber membranes for air filtration. These membranes mimic the intricate structure of spider webs, known for their ability to catch prey efficiently. The biomimetic approach, combined with an ordered structure, results in a material with a multistage pore architecture that exhibits high porosity, small pore size, and low filtration resistance. Such characteristics are ideal for intercepting a wide range of pollutants, including solid aerosols, liquid droplets, and viruses, through a screening effect that does not depend on electrostatic assistance. The research explored the influence of various metal chlorides on the formation of the nano-spiderweb fibers. The addition of metal chlorides to the spinning solution affects its conductivity, surface tension, and viscosity. This modification enhances electrostatic repulsion on the jet surface during the electrospinning process, leading to fiber splitting and the production of the desired nano-spiderweb structure. The resulting ordered PAN/nano spiderweb composite membranes boast not only exceptional hydrophobicity and air permeability but also a high filtration efficiency for particulate matter (99.8060% for PM_0.3_) and a superior bacterial filtration efficiency (>95%), which has great potential for personal protective equipment and air purification applications.

Table 2 provides typical breathable membranes prepared with different materials and their applications in electronics, medical, textile, packaging, and environmental fields. The addition of antimicrobial materials (e.g., thymol, ZnO nanoparticles, and cinnamaldehyde) and the use of electrospinning, blown film extrusion, and other methods of preparation make the breathable membranes able to be used in the medical field and packaging applications. By adding some heat-insulating materials (e.g., silica and silver nanoparticles), and using electrospinning, blending technology, etc., the breathable membranes can be used in the textile field. Adding some conductive materials or piezoelectric materials (e.g., multiwalled carbon nanotubes and PVDF) can be used in the field of electronics by electrospinning. The addition of hydrophobic agents (e.g., hydrophobic SiO_2_) to prepare a protective layer using the water surface spreading method can be used in the energy sector. Adding materials with adsorption properties (e.g., SBE-βCD) and preparing a special pore structure by electrospinning have a filtering effect on pollutants in the atmosphere and can be used in the environmental field.

## 4. Conclusions and Prospects

Breathable membranes are widely used in human daily life because of their waterproof and breathable properties. In this review, four main preparation methods of breathable membranes are introduced, including blown film extrusion, cast film extrusion, phase separation, and electrospinning. The research progress in the application of breathable membranes spans several fields, each benefiting from the membranes’ specific properties.

In the medical field, the addition of some materials with antimicrobial properties enables breathable membranes to be used in surgical drapes, gowns, and wound dressings, which provide a barrier against microorganisms while allowing the skin to breathe. In the electronics field, multifunctional electronic membranes prepared by adding materials with conductive or piezoelectric can be used in wearable electronic devices that can perform sensing, signal amplification, processing, and communication tasks for applications such as medical detection and monitoring. For textiles, breathable membranes are employed in outdoor and sports apparel to offer protection against the elements while maintaining comfort through moisture management. In addition, the heat retention and warmth of textiles are improved by incorporating several materials with heat-insulating properties into the membranes. In packaging, breathable membranes maintain the freshness of perishable goods by allowing gases to escape while blocking external contaminants. The inclusion of some antimicrobial agents can extend the shelf life of food products. In energy, breathable membranes are applied in battery separators and fuel cell components to improve efficiency and longevity. For the environment, the incorporation of functional groups with adsorptive properties in air filtration systems allows breathable membranes to remove airborne contaminants while allowing adequate airflow.

The ever-evolving demands of complex societal production and living environments have catalyzed significant advancements in the research and development of breathable membranes. These advancements are not only broadening the application spectrum of breathable membranes but also emphasizing the critical need for environmental stewardship and sustainability in their development and use. In the future, the focus on green, eco-friendly, renewable, and biodegradable bio-based materials for breathable membranes is expected to intensify. This shift is driven by a collective aspiration for sustainable development, aiming to mitigate environmental impacts through the adoption of green materials and manufacturing processes. Furthermore, enhancing the recyclability and biodegradability of breathable membranes will emerge as a pivotal area of development, ensuring these materials contribute positively to environmental sustainability. Technological innovation and process optimization will continue to propel the field of breathable membranes forward. The development of new materials with significant application value will play a crucial role in supporting sustainable growth in this domain. Expanding the applications of breathable membranes in building materials for insulation and waterproofing, cosmetics and personal care products for enhanced skin comfort, and medical devices including respiratory machines and artificial skin, underscores the versatile potential of these materials. The demand for multifunctional and high-performance breathable membranes is poised to drive the use of high-performance polymers. The future may see the emergence of intelligent membranes equipped with self-awareness, self-cleaning, self-adaptation, and self-repair capabilities, opening up new possibilities in intelligent electronics, packaging, medical, and other fields. Such developments will pave the way for breathable membranes that not only prioritize environmental friendliness but also offer broader, more versatile applications, ultimately contributing to a more comfortable, healthy, and convenient life for people around the globe.

## Figures and Tables

**Figure 1 polymers-16-01686-f001:**
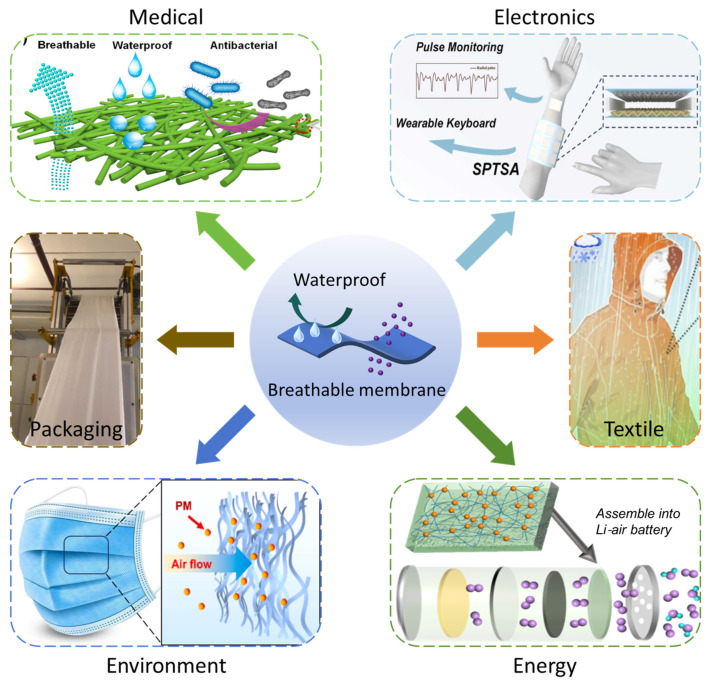
The recent advances of breathable membranes in medical [8], electronics [10], textile [12], packaging [14], energy [18], and environmental fields [21].

**Figure 2 polymers-16-01686-f002:**
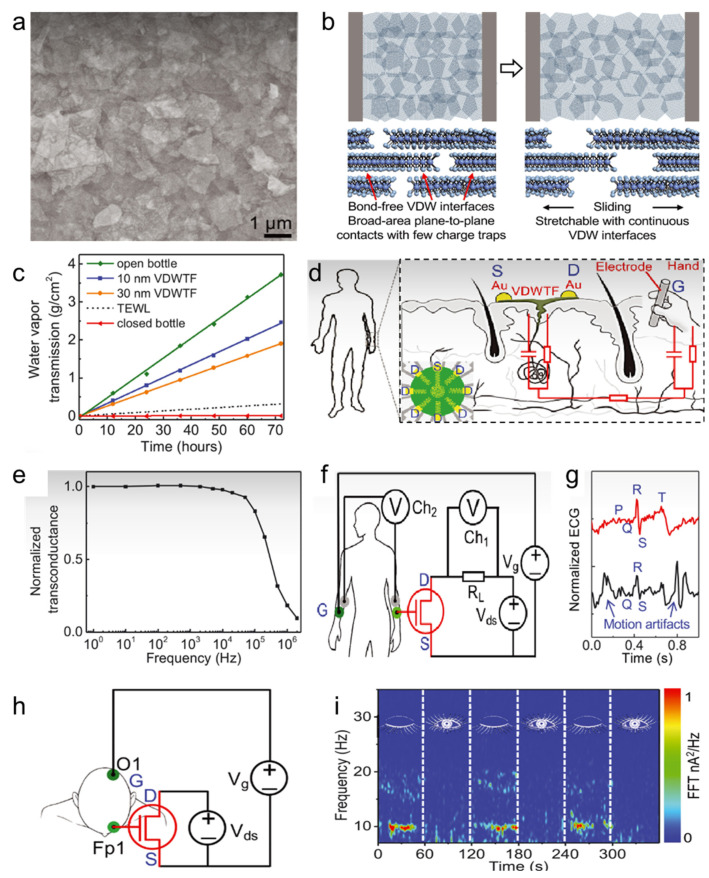
(**a**) SEM morphology, (**b**) schematic before and after stretching, (**c**) air permeability, (**d**) skin-gate VDWTF transistor, (**e**) normalized transconductance at different frequencies, (**f**) ECG measurements, (**g**) PQRST waveforms, (**h**) EEG measurements, and (**i**) cyclic closed-eye and open-eye EEGs of a VDWTF [1].

**Figure 3 polymers-16-01686-f003:**
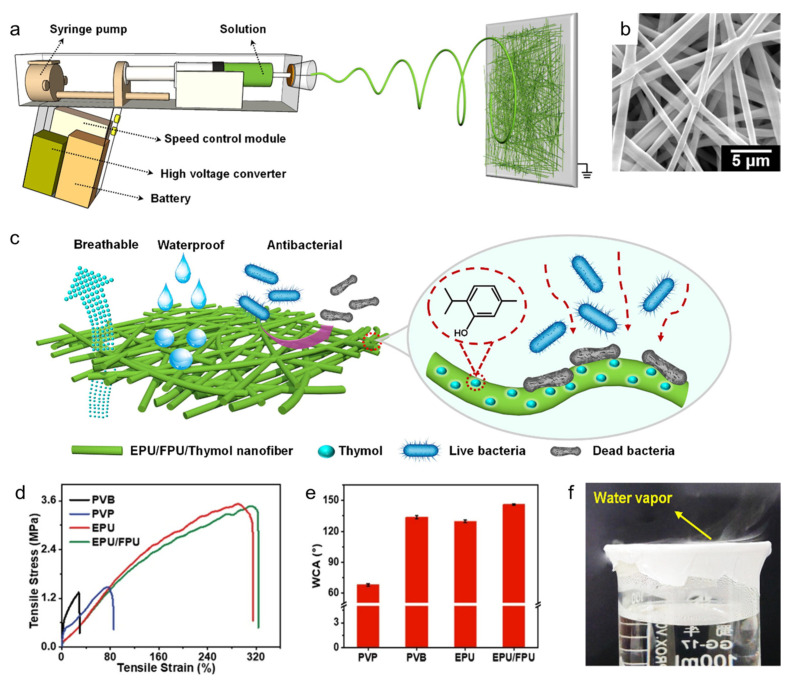
(**a**) Preparation, (**b**) waterproof, breathable, and antibacterial properties, (**c**) SEM image, (**d**) stress–strain curve, (**e**) hydrophobicity, and (**f**) breathability of EPU/FPU/thymol nanofiber breathable membrane [8] for wound dressings.

**Figure 4 polymers-16-01686-f004:**
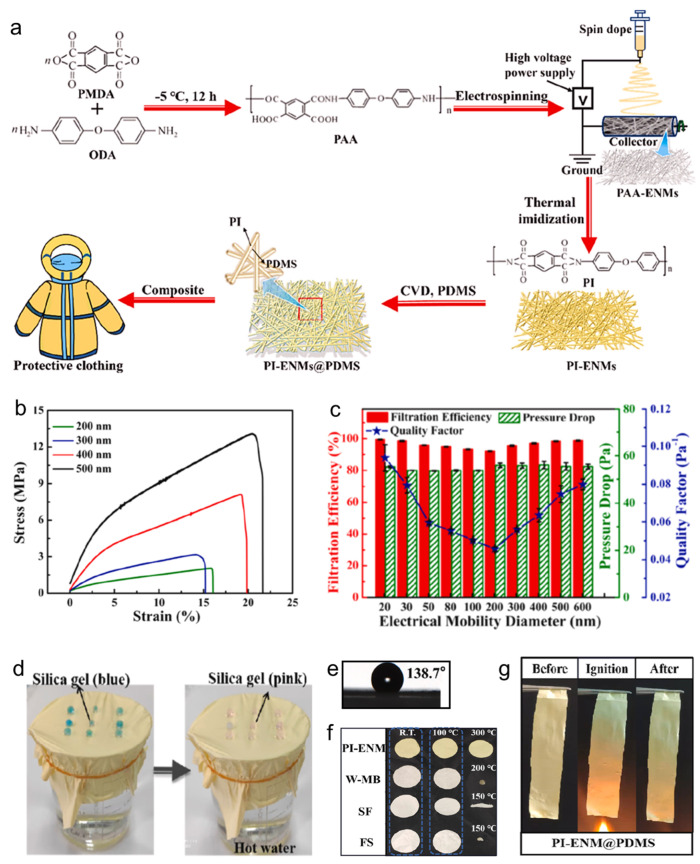
(**a**) Preparation, (**b**) mechanical properties, (**c**) filtration, (**d**) air permeability, (**e**) water resistance, (**f**) thermal stability, and (**g**) flame retardancy of PI-ENMs [7].

**Figure 5 polymers-16-01686-f005:**
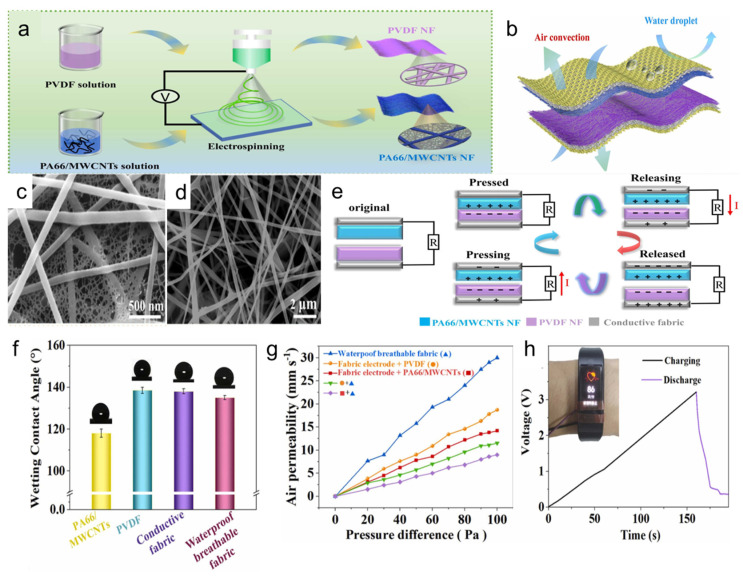
(**a**) Preparation, (**b**) waterproof and breathable schematic diagram, (**c**) SEM image of positive electrode film, (**d**) SEM image of negative electrode film, (**e**) working principle, (**f**) hydrophobicity, (**g**) breathability, and (**h**) application of breathable and waterproof triboelectric nanogenerator [9].

**Figure 6 polymers-16-01686-f006:**
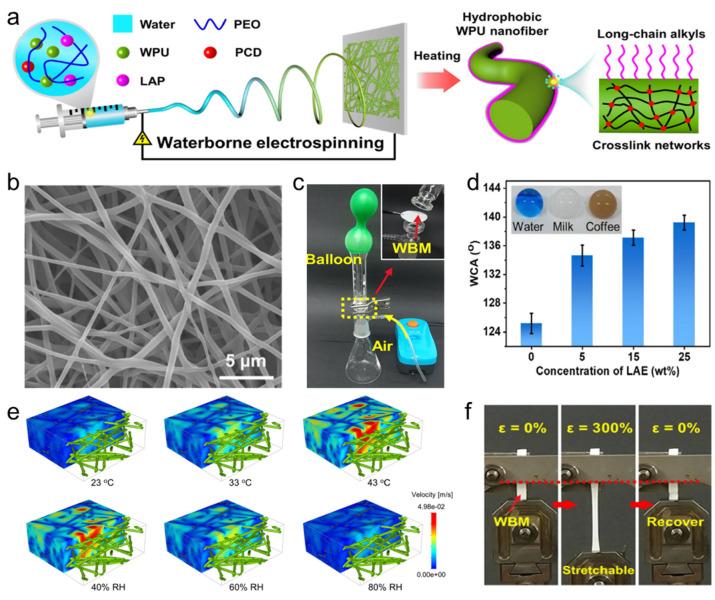
(**a**) Preparation, (**b**) SEM image, (**c**) air permeability, (**d**) water repellency, (**e**) water vapor permeability at different temperatures and relative humidities, and (**f**) mechanical properties of waterborne fluorine-free nanofiber membranes [11] for protective textiles.

**Figure 7 polymers-16-01686-f007:**
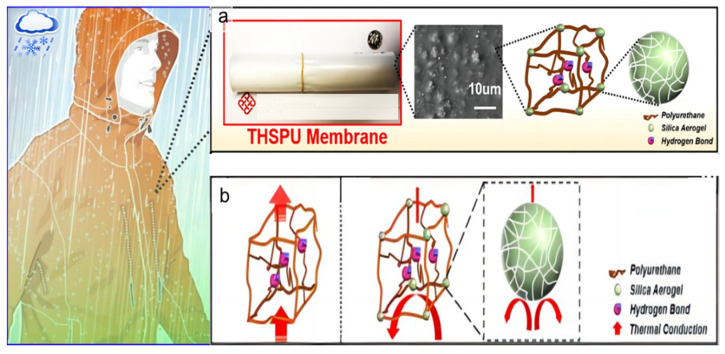
(**a**) Preparation and (**b**) heat retention of polyurethane-based breathable membrane [12].

**Figure 8 polymers-16-01686-f008:**
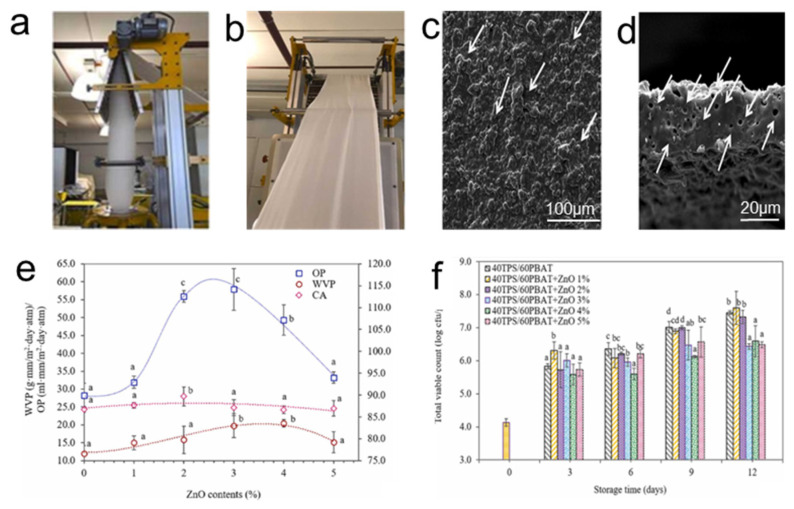
(**a**) Preparation device, (**b**) appearance, (**c**) surface SEM image, (**d**) cross-section SEM image, (**e**) water vapor permeability (WVP), oxygen permeability (OP), and contact angle (CA) of PBAT/TPS/ZnO films. (**f**) Total viable count (TVC) of pork packaged using ZnO/PBAT/TPS films stored at 4 °C [14]. (White arrows in subfigures (**c**,**d**) indicate merge granules and pores in surface and cross-section microstructures, respectively. The letters in subfigures (**e**,**f**) in caption above the error bars refer to the level of statistical significance).

**Figure 9 polymers-16-01686-f009:**
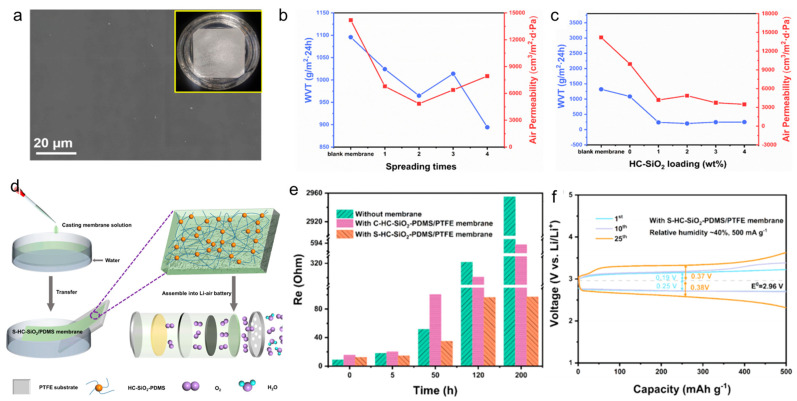
(**a**) SEM image, (**b**) effect of spreading time on water vapor and permeability, (**c**) effect of HC-SiO_2_ mass fraction on water vapor and permeability, (**d**) assembly diagram of lithium−air battery, (**e**) relationship between Ohmic resistance (Re) and time, and (**f**) discharge/charge curve of PDMS/PTFE film [18].

**Figure 10 polymers-16-01686-f010:**
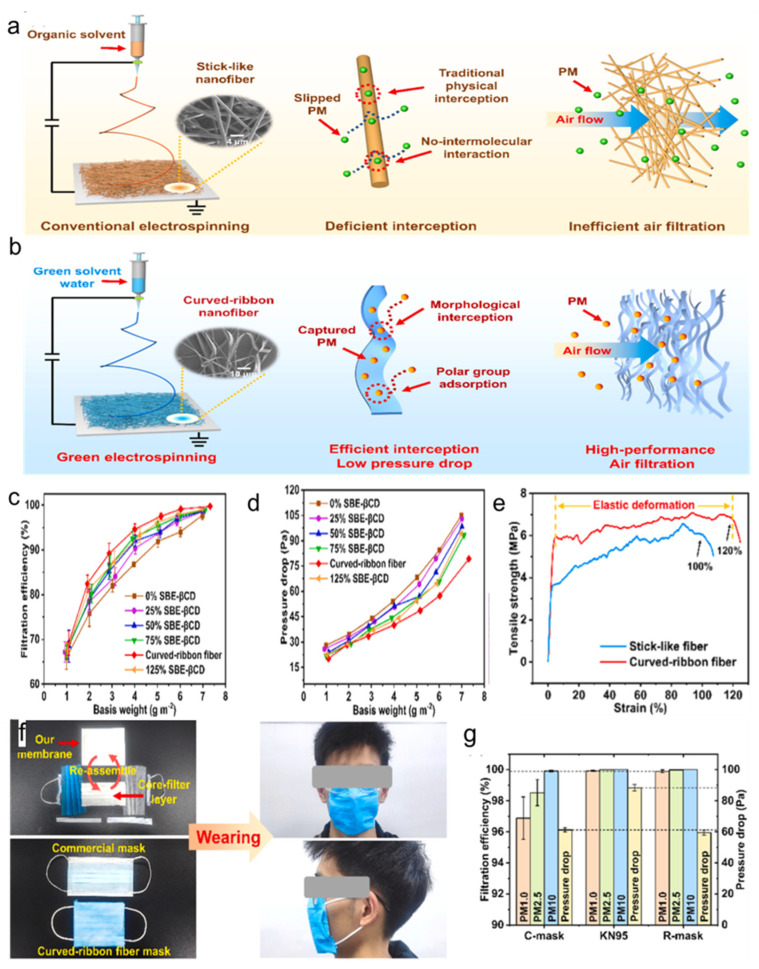
Breathable and environmentally friendly SBE−βCD/PVA nanofiber membranes [21] for air filtration with (**a**) rod nanofibers, (**b**) curved-ribbon nanofibers, (**c**) filtration efficiency, (**d**) pressure drop performance, (**e**) mechanical properties, (**f**) application in protective masks, and (**g**) filtration efficiency and pressure drop performance of different masks.

**Table 1 polymers-16-01686-t001:** Comparison of the preparation methods for breathable membranes.

Preparation Methods	Polymer State	Pore Formation Mechanism	Impact on the Environment	References
Blown film extrusion	Polymer melt	Stretching caused separation between the filled particles and the matrix	Almost no impact	[36]
Phase separation	Polymer solution	Nonsolvent-induced phase separation	Organic solvents may pollute the environment	[50]
Phase separation	Polymer solution	Thermally induced phase separation	Organic solvents may pollute the environment	[53]
Cast film extrusion	Polymer melt	Stretching caused separation between the filled particles and the matrix	Almost no impact	[72]
Cast film extrusion	Polymer melt	Stretching caused separation between the crystals and the amorphous polymer	Almost no impact	[42,73]
Electrospinning	Polymer solution	Accumulation of fibers	Organic solvents may pollute the environment	[63]

**Table 2 polymers-16-01686-t002:** Typical breathable membranes prepared with different materials and their applications.

Materials	Properties	Applications	References
Ethanol-soluble polyurethane/fluorinated polyurethane/thymol	Antibacterial	Medical	[8]
Acid/polyurethane/sodium periodate/dopamine/ethyl orthosilicate	Antibacterial and protective	Medical	[81]
Artemisia argyi oil/sodium alginate/polyvinyl alcohol	Antibacterial	Medical	[89]
Halloysite nanotubes/silver nanowires/polyurethane	Antibacterial	Medical	[85]
PA66/multiwalled carbon nanotubes/polyvinylidene fluoride	Conductivity	Electronics	[9]
Carboxylated multi-walled carbon nanotubes/poly(3,4-ethylenedioxythiophene)/poly(styrenesulfonate)/thermoplastic polyurethane	Sensitivity	Electronics	[74]
Polyaniline/poly (vinylidene fluoride)/cetyltrimethylammonium bromide	Sensitivity	Electronics	[75]
Waterborne polyurethane/polycarbodiimide/long chain alkyl polymer	Hydrophobicity	Textile	[11]
Thermoplastic polyurethane/silica	Heat retention	Textile	[12]
Silver nanoparticles/Wastepaper cellulose	Antibacterial and thermal insulation	Textile	[13]
Poly (butylene adipate-*co*-terephthalate)/thermoplastic starch/ZnO nanoparticles	Antibacterial and barrier	Packaging	[14]
PIM-1/polylactic acid	Antibacterial	Packaging	[15]
Poly (lactic acid)/ polycaprolactone/sodium chloride/poly (ethylene oxide)	Antibacterial	Packaging	[16]
Low density polyethylene/silver/ZnO	Barrier	Packaging	[112]
Polybutylene adipate/polybutylene succinate/linear low-density polyethylene	Antibacterial and barrier	Packaging	[113]
Hydrophobic SiO_2_ /silicalite-1/polydimethylsiloxane	Waterproof and selective	Energy	[18]
Poly (vinylidene fluoride-*co*-hexafluoropropylene)/dextrin-nanosponge	Waterproof and selective	Energy	[19]
Sodium sulphobutylether-β-cyclodextrin/polyvinyl alcohol	Filterability and adsorption	Environment	[21]
Polyacrylonitrile/nano-spiderweb composite nanofibre	Filterability	Environment	[22]
Halloysite nanotubes /ZnO nanoparticles/polycaprolactone	Filterability	Environment	[129]

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
