# Peer review of "Progress in the Preparation and Application of Breathable Membranes"

_polymers, 2024, doi:10.3390/polym16121686_

Round 1

Reviewer 1 Report

Comments and Suggestions for Authors

Review for the paper “Progress in the preparation and application of breathable membranes” by Tingshuai Luo and co-authors submitted to “Polymers”.

The impermeability of waterproof and breathable layers to water is balanced by their permeability to water vapor and air. The combination of waterproof and breathable membranes offers a unique set of properties, including resistance to liquid water penetration and the ability to transmit water vapor, which are crucial in addressing challenges related to health, resources, and energy. The versatility of these materials allows their adaptation to a wide range of applications, from healthcare to environmental protection.

The authors provided a comprehensive overview of the preparation and application of breathable membranes in a range of industries, including electronics, medicine, textiles, packaging, energy, and the environment. This paper was written in an engaging and informative style and included recent findings. However, some updates and revisions are necessary before the final acceptance.

Recommendations.

L 36-40. These sentences require relevant citations.

The rationale for this review article should be highlighted by the authors. Why the preparation and application of breathable membranes require further exploring if there are a number of previous and recent reviews in this topic (references 1–24)? The authors should formulate the scientific and/or practical problems that are needed to be solved.

Section 2. The authors should pay more attention to advantages and disadvantages of each method in terms of their impacts on the environment. The authors should update Table 1 with this information.

L 206-216. This text is more relevant for the “Medical” section.

Section 3.2. L 289. kg m−2d−1 the units should be formatted in the upper register as kg m−2 d−1

The authors should consider the following papers to update the text by recent advances.

Cai, L., Xu, L., Si, Y., Yu, J., & Ding, B. (2021). Autoclavable, Breathable, and Waterproof Membranes Tailored by Ternary Nanofibers for Reusable Medical Protective Applications. ACS Applied Polymer Materials, 4(1), 556-564.

Yao, Y., Guo, Y., Li, X., Yu, J., & Ding, B. (2021). Asymmetric wettable, waterproof, and breathable nanofibrous membranes for wound dressings. ACS Applied Bio Materials, 4(4), 3287-3293.

Zhao, Y., Wang, T., Li, Y., Zhao, Z., Xue, J., & Wang, Q. (2024). Fabrication of Breathable Multifunctional On-Skin Electronics Based on Tunable Track-Etched Membranes. ACS Applied Electronic Materials.

L 406. H2O should be formatted in the lower register as H2O

Section 3.4.

The authors should include information from the following papers to update their review:

Préfol, T., Gain, O., Sudre, G., Gouanvé, F., & Espuche, E. (2021). Development of breathable pebax®/peg films for optimization of the shelf-life of fresh agri-food products. Membranes, 11(9), 692.

Alisiyonak, O., Lavitskaya, A., Khoroshko, L., Kozlovskiy, A. L., et al., (2023). Breathable Films with Self-Cleaning and Antibacterial Surfaces Based on TiO2-Functionalized PET Membranes. Membranes, 13(8), 733.

Yun, X., Lu, H., Zhou, Z., Yuan, S., Wang, Y., & Dong, T. (2023). Fabrication and design of poly (llactic acid) membrane for passive MAP packaging of Brassica chinensis L. Journal of Food Science, 88(4), 1640-1653.

Section 3.5.

The authors should include information from the following papers to update their review:

Song, H., Xu, S., Li, Y., Dai, J., Gong, A., et al. (2018). Hierarchically porous, ultrathick,“breathable” woodderived cathode for lithiumoxygen batteries. Advanced Energy Materials, 8(4), 1701203.

Terutsuki, D., Okuyama, K., Zhang, H., Abe, H., & Nishizawa, M. (2022). Water-proof anti-drying enzymatic O2 cathode for bioelectric skin patch. Journal of Power Sources, 546, 231945.

Specific remarks.

L 15. Consider replacing “have wide range” with “have a wide range”

L 18. Consider replacing “in physical properties” with “of physical properties”

L 36. Consider replacing “demand of comfort” with “demand for comfort”

L 49. Consider replacing “breathable membrane” with “breathable membranes”

L 112. Consider replacing “the temperature of the casting roll control” with “the temperature of the casting roll controls”

L 116. Consider replacing “various application” with “various applications”

L 124. Consider replacing “The structure and property” with “The structure and properties”

L 179. Consider replacing “medical” with “medicine”

L 236. Consider replacing “materials are closely oscillated” with “materials closely oscillates”

L 279. Consider replacing “Yue et al. [9] involves” with “Yue et al. [9] involve”

L 435. Consider replacing “significantly contribute” with “significantly contributes”

L 626. Consider replacing “airflow making air permeability reduced” with “airflow reducing air permeability”

L 627. Consider replacing “worth investigating on how” with “worth investigating how”

L 652. Consider replacing “and its applications” with “and their applications”

L 679. Consider replacing “In future” with “In the future”

Comments on the Quality of English Language

Minor revision

Reviewer 2 Report

Comments and Suggestions for Authors

​Gas-permeable (breatheable) membranes are used to ensure air permeability, to separate valuable gases, separate gas mixtures during scientific research, in medicine, etc. The use and development of ideas about such membranes is extremely important. For breatheable membranes, the authors identify six main directions, which are presented in Figure 1. However, from the presented figure, the achievements indicated by the caption are not entirely clear. The authors consider several methods for forming membranes, including the melt method (extrusion), NIPS and TIPS methods, and electrospinning. Traditional industrial polymers such as PAN, PU, etc. are considered as polymers. After a brief consideration of methods for forming breatheable membranes, the main directions of use of such membranes are considered. When considering breatheable membranes for electronic devices, the question arises about the feasibility of such membranes. There is no conclusion about the requirements for such membranes. For medical materials, non-flammability is noted; this property of medical materials is doubtful. For textile membranes for separating emulsions (oil-water systems), it is not clear why gas permeability is needed? For packaging membranes, the question of the price of the PIM-1/PLA system arises. How much do such materials cost? Is it profitable to pack food, etc. in them? How can such membranes be disposed of later?

The authors mention various methods for forming membranes, which can be identified in a list of keywords. Now it only has 4 words.

Line 28. "liquid water" I suggest removing "liquid"

Lines 169-171. It’s not clear what this part and the following table 1 are for?

Line 519. "increases., due" - typo

Table 2. It is difficult to understand how relevant this table is?! Some links are already given in the text with mention of membranes.

The conclusions generally highlight the material presented in the review, but in my opinion the emphasis is not placed entirely accurately. The authors focus on the properties and application of membranes, but do not provide specific characteristics on which to rely when developing and using such materials. It seems to me that it would be more interesting if the authors drew conclusions about the optimal conditions for the formation of such membranes and polymers (composite systems) from which multifunctional breatheable membranes can be obtained.

Reviewer 3 Report

Comments and Suggestions for Authors

The current review is very interesting and worthwhile publishing. Below some minor comments:

Section 1: Please add Chem. Soc. Rev., 2022, 51, 4537-4582. One needs in this review also to highlight the chemistry layer somewhat more (see also further).

Section 2: add some graphs of the working principle of these manufacturing techniques.

Most important comment for Section 2: each subsection is relatively fine as such but a comparison of the techniques is missing. So either add a table or a short final section on advantages and disadvantage for the general reader. (so more than the current table which is still per “entry”).

L 168. A disclaimer can be made on the functionality level at the molecular level. So it is not only size of the micropores but also the type of functional groups on the surface of the electrospun membrane. In the case of water one needs to look at OH groups. Please make a disclaimer that the reaction conditions of the electrospinning matter as well. The most advanced example here that perhaps could be mentioned is done for molecular-based water passage for silica-based electrospun membranes (Nat. Mater. 2021 20, 1422).

Application part: I suggest to shorten it as often only 1 contribution is highlighted in full detail. Also a somewhat better link to the subplots could be worthwhile.

Author Response

请参阅附件

Round 2

Reviewer 2 Report

Comments and Suggestions for Authors

The keyword "Preparation" is not quite suitable, in my opinion, it is better to replace it. Also, for review, the entire list can be expanded by adding 2-3 more keywords.

Figure 1, in my opinion, remains uninformative and does not evoke associations for a number of sectors! It is not clear what recent achievements have been made in the direction of breathable membrane.

Figure 2 is of low quality. I suggest that this figure be either deleted or corrected, designing it in the same style.

Lines 186-187. "further expand the application of breathable membranes is the future of the promising direction of development. It is a very promising direction for the future." - I recommend correcting this part.

I don’t understand the meaning of Table 1. What do the authors think the reader should get from it and what conclusions should be drawn?

The quality of most of the review figures needs to be improved.
